# Structure of the herpes simplex virus type 2 C-capsid with capsid-vertex-specific component

Jialing Wang[1,2], Shuai Yuan[1], Dongjie Zhu [1], Hao Tang[3], Nan Wang[1,2], Wenyuan Chen[3], Qiang Gao[1,4], Yuhua Li[5], Junzhi Wang[5], Hongrong Liu[3], Xinzheng Zhang[1,2], Zihe Rao[1,2,6,7,8] & Xiangxi Wang [1,2]

Herpes simplex viruses (HSVs) cause human oral and genital ulcer diseases. Patients with HSV-2 have a higher risk of acquiring a human immunodeficiency virus infection. HSV-2 is a member of the *α-herpesvirinae* subfamily that together with the *β-* and *γ-herpesvirinae* subfamilies forms the *Herpesviridae* family. Here, we report the cryo-electron microscopy structure of the HSV-2 C-capsid with capsid-vertex-specific component (CVSC) that was determined at 3.75 Å using a block-based reconstruction strategy. We present atomic models of multiple conformers for the capsid proteins (VP5, VP23, VP19C, and VP26) and CVSC. Comparison of the HSV-2 homologs yields information about structural similarities and differences between the three herpesviruses sub-families and we identify α-herpesvirus-specific structural features. The hetero-pentameric CVSC, consisting of a UL17 monomer, a UL25 dimer and a UL36 dimer, is bound tightly by a five-helix bundle that forms extensive networks of subunit contacts with surrounding capsid proteins, which reinforce capsid stability.

[1] National Laboratory of Macromolecules, Institute of Biophysics, Chinese Academy of Science, Beijing 100101, China. [2] University of Chinese Academy of Sciences, Beijing 100049, China. [3] College of Physics and Information Science, Synergetic Innovation Center for Quantum Effects and Applications, Key Laboratory of Low-dimensional Quantum Structures, and Quantum Control of the Ministry of Education, Hunan Normal University, Changsha 410008, China. [4] Sinovac Biotech Co., Ltd, Beijing 100085, China. [5] National Institutes for Food and Drug Control, No. 2, Tiantanxili, Beijing 100050, China. [6] Shanghai Institute for Advanced Immunochemical Studies, ShanghaiTech University, Shanghai 201210, China. [7] State Key Laboratory of Medicinal Chemical Biology, Nankai University, Tianjin 300353, China. [8] Laboratory of Structural Biology, Tsinghua University, Beijing 100084, China. These authors contributed equally: Jialing Wang, Shuai Yuan, Dongjie Zhu.  Correspondence and requests for materials should be addressed to H.L. (email: hrliu@hunnu.edu.cn) or to X.Z. (email: xzzhang@ibp.ac.cn) or to Z.R. (email: raozh@tsinghua.edu.cn) or to X.W. (email: xiangxi@ibp.ac.cn)

Herpesviruses constitute a large family of dsDNA viruses, which are the causative agents of a range of diseases, including oral and genital blisters (herpes simplex viruses, HSV-1, and HSV-2), congenital disorders in immune-compromised individuals (Human cytomegalovirus, HCMV) and cancers (Epstein–Barr virus (EBV) and Kaposi sarcoma herpesvirus (KSHV))[1]. Based on their biological properties and genome sequences, the family *Herpesviridae* is divided into three subfamilies: *α-*, *β-*, and *γ-herpesvirinae*[2] and the nine known human herpesviruses span all three subfamilies. HSV-1, HSV-2, and varicella-zoster virus (VZV), which belong to the *α-herpesvirinae* subfamily, are present in a high proportion of adult populations globally and can establish lifelong latent infections within the peripheral nervous system. In contrast, genetically modified forms of HSV-1/HSV-2, designed to replicate specifically in tumor cells and lyse tumor-specific cells, have been used therapeutically[3]. A clear understanding of the structure and function of the structural proteins of herpesviruses could help assist in the design of anti-viral agents as well as improve their utility and efficiency as a therapeutic agent for treating tumors.

HSV has a characteristic particle structure comprising a DNA-filled capsid (~125 nm diameter), a proteinaceous tegument layer, and a lipid envelope[4]. In addition to protecting the genome, the capsid functions in retrograde transport in the host cell, release of the viral genome into the nucleus of the host cell, and mediation of the egress of nascent capsid from the cell nucleus[5,6]. Three assembly intermediate capsids termed A-, B-, and C-capsids can be isolated from lysates of infected cells[7]. A-capsids are empty and result from abortive DNA packing, while B-capsids comprise a core including scaffold proteins. Whether B-capsids are abortive forms or assembly intermediates is still debated[8,9]. The third type of capsid, C-capsid, is fully packed with the DNA genome and matures into an infectious virion. All three types of capsids have mature angular shells (triangulation number $T = 16$), composed primarily of 955 copies of the 150 kDa major capsid protein, VP5, arranged as 150 hexons and 11 pentons[10]. The dodecameric UL6 portal complex occupies the 12th vertex, breaking the local and icosahedral symmetry. 320 copies of the "triplex" comprising two copies of VP23 and one copy of VP19C connect adjacent capsomers[11] while 900 copies of VP26 cap the outer surface of hexons but not pentons[12]. Compared to A- and B- capsids, C-capsids have significantly higher occupancy of the capsid-vertex-specific component (CVSC), comprising UL17, UL25, and the putative UL36 proteins[13] disposed around the exterior of each of the capsid vertices and implicated in DNA packing and capsid maturation[14]. The DNA genome is under considerable pressure within the capsid as shown by cryo-EM micrographs of HSV-1 C-capsids, where the inter-duplex spacing is measured at ~26 Å, a value comparable to that found in dsDNA bacteriophages and near the theoretical limit for close packing of DNA duplexes[15]. The CVSC reinforce the capsid structure to withstand high internal genome pressures and may signal the completion of DNA packing, thereby initiating nuclear egress[14,16]. The CVSC homolog has also been identified in γ-herpesvirus KSHV, but with a relatively low occupancy in contrast to the occupancy of CVSC in α-herpesviruses[17]. However, the CVSC in the β-herpesvirus HCMV bears neither compositional nor structural similarities to those found in the α- and γ-herpesviruses, whilst the β-herpesvirus-specific tegument protein pp150, an evolutionary substitute for CVSC, forms a global net that allows DNA-filled C-capsids to cope with the pressure of its large genome[18]. The necessities to mediate binding of divergent tegument proteins may facilitate the most external components of the capsid (e.g., VP26 and CVSC) to diversify across the different subfamilies of herpesviruses.

Persistent efforts in the past two decades have pushed the resolution limit of cryo-EM analysis of herpesvirus capsids from 15 Å to 6 Å[11,19–21] and further down to the most recent resolutions reported for the structures of the β-herpesvirus HCMV and the γ-herpesvirus KSHV at 3.9 Å and 4.2 Å, respectively[17,18]. However neither of these structures reveal structural information on auxiliary tegument proteins that bind exclusively to pentons and peripentonal triplexes to reinforce particle stability. Our recent determination of the structure of the HSV-2 B-capsid[22], together with the structure of the HSV-1 C-capsid[23], expands our understanding of the drivers of assembly and the basis of stability of the capsid in α-herpesviruses. Using our recently-developed block-based reconstruction method[24] we have determined the cryo-EM structure of HSV-2 C-capsid at 3.75 Å and have built atomic models for the capsid and CVSC. This structural information, together with recently-reported four near-atomic structures spanning all three herpesvirus sub-families, allows us to identify α-herpesvirus-specific structural features, providing insights into the early evolution of herpesviruses.

## Results

**The block-based reconstruction and overall structure.** To alleviate the Ewald sphere effects caused by the large size of virions (>200 nm in diameter), we analyzed detergent-treated HSV-2 (*MS* strain) capsids instead of the intact virions for cryo-EM imaging (Methods). Cryo-EM micrographs of purified HSV-2 C-capsids were recorded using an FEI Titan Krios electron microscope equipped with a Falcon detector (Methods). A total of ~50,000 particles were selected from the cryo-EM images and subjected to two-dimensional (2D) alignment and three-dimensional (3D) reconstruction with icosahedral symmetry imposed in Relion[25], which yielded a reconstruction of 4.2 Å resolution. The 4.2 Å resolution map reveals the icosahedrally ordered components of the virion, including pentons, 3 types of hexons (P, peripentonal; E, edge; C, center) with the hexameric rings formed by VP26s, 320 quasi-equivalent triplexes (Ta-Tf) and 12 pentagram-shape CVSC densities (Fig. 1a). Each asymmetric unit contains a C-Hex, P-Hex, one-half of an E-Hex, one-fifth of a Pen, 15 copies of VP26, $5^{1/3}$ triplexes and one CVSC (Fig. 1b).

There are two bottlenecks that limit the cryo-EM resolution (~4 Å) of this 1250 Å diameter capsid. One is the complex architecture that does not strictly conform to the icosahedral symmetry; the other is the gradient in defocus through the capsid. To overcome these, we developed a block-based reconstruction method[24]. Briefly, any large object with a big defocus gradient can be split into several smaller blocks so that the defocus gradient on each block is much less than that of the whole object and each block can be reconstructed separately with its local mean defocus (Supplementary Fig. 1). In our case, icosahedral orientation and center parameters of each particle image determined by Relion[25] were used to guide extraction of components of all four block regions (Pen-, P-Hex-, E-Hex-, C-Hex-blocks, ~50% bigger than each capsomers) and all four blocks were refined and reconstructed separately with their local mean defoci (Supplementary Fig. 1). After refinement and reconstruction of the four blocks in Relion[25], the resolution of maps for Pen-, P-Hex-, E-Hex-, and C-Hex-blocks were further improved to 4.0 Å, 3.75 Å, 3.72 Å, and 3.71 Å respectively, as determined by Gold standard Fourier shell correlation at the 0.143 threshold (Supplementary Fig. 2). The four blocks can be combined into a complete map that contains an intact asymmetric unit yielding a final resolution of 3.75 Å for the HSV-2 C-capsid (Fig. 1a-c and Supplementary Fig. 2). The resulting density map features well-resolved side chains consistent with this resolution (Fig. 1b and Supplementary Table 1),

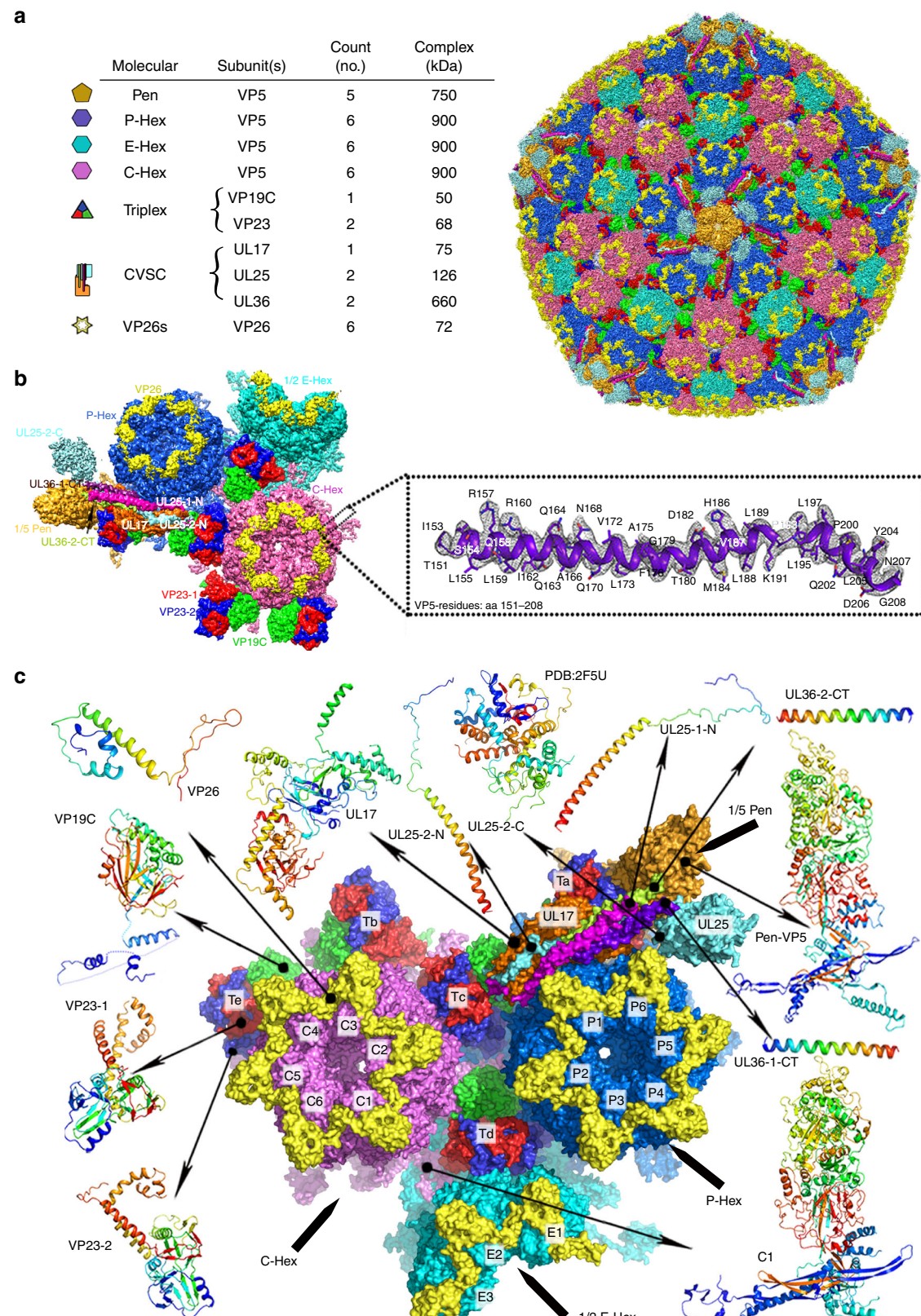

**Fig. 1** Architecture of the HSV-2 C-capsid. **a** Surface representation of HSV-2 C-capsid. The table lists the major capsid elements as identified by color in the capsid. **b** Cryo-EM map of an asymmetric unit and local electron density maps are shown. The inset shows the density map (mesh) and atomic model of VP5 which illustrate side chain features. Residues with side chains are labeled, aa denotes amino acids. **c** Ribbon diagram of the atomic model of an asymmetric unit. The triplex consists of two VP23 conformers (denoted as VP23-1 and VP23-2) and one copy of VP19C. UL36-CT denotes UL36 C-terminal helix. The CVSC comprises a UL17 monomer, two UL25 conformers (denoted as UL25-1 and UL25-2) and two UL36 conformers (denoted as UL36-1 and UL36-2). Rainbow ribbon models show individual proteins and conformers (blue N terminus through green and yellow to red C terminus)

allowing atomic models for all four capsid proteins (VP5, VP26, VP23, and VP19C) and the CVSC to be built in. The atomic models of an asymmetric unit (ASU) include 16 copies of VP5, 15 copies of VP26 making up a C-Hex, a P-Hex, one-half of an E-Hex and one-fifth of a Pen, 5 triplexes (Ta, Tb, Tc, Td, Te) and one CVSC (Fig. 1c).

**α-Herpesvirus-specific structural features**. α-Herpesviruses are commonly defined by their ability to establish a latent infection in neurons, which is a distinctive feature not found in β- and γ-herpesviruses. Upon infection of sensory nerves innervating the skin or mucosa, the C-capsids undergo long-range retrograde axonal transport to the neuronal cell body, during which capsid-bound tegument proteins are involved[26]. Additionally, long distance transport involving mechanical forces generated by molecular motors requires high stability of C-capsids. The cryo-EM structure of HSV-2 C-capsid reveals an icosahedral assembly of ~1250 Å in diameter, similar to HSV-1, HCMV, and KSHV C-capsids and HSV-2 B-capsid (Supplementary Fig. 3). The pressure of tens of atmospheres generated by the packing of the genome[27] do not expand the C-capsid, suggesting C-capsids possess exceptional mechanical and structural stability/rigidity, which is consistent with previous results unveiled with fluid atomic force microscopy[28]. Surrounding each capsid vertex, HSV-1 and HSV-2 C-capsids present pentagram-shaped densities of CVSC (parts of CVSC labeled by pink elliptical lines are flexible in HSV-2) (Fig. 2a). The density values of CVSC when compared with those of its underlying triplexes (Ta and Tc) are quite similar (the ratio of the maximum density of the CVSC to the maximum density of its underlying triplexes is 0.95, assuming the latter density to reflect 100% occupancy), suggesting nearly full occupancy of CVSC. This is in contrast to the low occupancy of CVSC in the virion of KSHV[17] and HSV-2 B-capsid[22] (Supplementary Fig. 3). Nevertheless, in contrast to HCMV, where the global binding of the β-herpesvirus-specific tegument protein pp150 produces a rather smooth surface, the vertices are slightly raised in HSV-1, HSV-2, and KSHV, giving the viruses more angular appearances (Fig. 2a and Supplementary Fig. 3). Except for the capsid-associated tegument proteins, the C-capsids from all three subfamilies of herpesviruses have the same protein compositions: VP5, VP23, VP19C, and VP26. Amongst these capsid proteins, VP26 is the most divergent both structurally and functionally (e.g., HSV-2 VP26 is ~50% larger than its homolog SCP in HCMV). In α-herpesviruses, 6 copies of VP26 form a ring-shaped structure by end-to-end interactions crowning only hexons, while SCPs bind both hexons and pentons in both β- and γ-herpesviruses (Fig. 2a). VP19C which, together with 2 copies of VP23 makes up the "quasi-trimeric" triplex, is about 40% longer than those of VP19C homologs in both β- and γ-herpesviruses, whereas the sequences of VP23 are conserved in length and similarity across the three herpesvirus subfamilies. Unsurprisingly, VP19C in HSV-1 and HSV-2 bears an extra 55-amino acid insertion-arm domain (disordered in HSV-2 due to the lack of stabilization/protection by tegument proteins), sitting on the head of the two VP23 conformers and a longer (~100-amino acid) N-anchor (disordered in HSV-1), penetrating the capsid floor when compared with the structures of VP19C homologs from HCMV and KSHV (Fig. 2b). The necessity to interact with tegument proteins located outside the capsid and accommodate genomes of various sizes inside the capsid probably require and cause alterations in VP19C on both its outer and inner sides. In order to accommodate the 235 kb genome (~50% larger than that of HSV-1) within the similar-size capsid successfully, HCMV has probably evolved distinctive strategies including shortening of its N-anchor of VP19C from 105 to 44 residues to increase the inner

space and compressing the genome into hexon channels to take full advantage of the inner space[29].

VP5, the most conserved capsid protein across the three subfamilies, is also folded into seven domains: upper, buttress, helix-hairpin, channel, Johnson fold, dimerization and N-lasso (Supplementary Fig. 4). Like other herpesvirus capsids[22], none of the 16 copies of VP5 in an asymmetric unit is identical and they can be grouped into 4 sub-classes: typical hexon-C1, P1, P6, and typical penton-Pen1 (subunits from C-Hex, P-Hex, E-Hex, and Pen denoted as C1 to C6, P1 to P6, E1 to E6, and Pen1 to Pen5, respectively) (Fig. 1). The capsid shell is assembled via extensive interactions of the lower sections of VP5, including both intracapsomer and intercapsomer interactions (Fig. 2b). Interestingly, in α-herpesviruses a set of five helix-pairs comprising two long αN helices from the dimerization domain of Pen1 and P6 is observed at the inner surface beneath the penton, forming a new type of quasi-equivalent twofold interaction (Fig. 2b). In addition, P6 as well as penton VP5s adopt α-herpesvirus-specific configurations at the dimerization domain, that refolds from the helix-turn-helix structure (observed in P1 and typical-hexon VP5s) into a single long helix (Fig. 2b). However, these conformational changes in β-herpesviruses are not very significant at the dimerization domain and the local quasi-equivalent twofold interactions between the penton VP5s and P6s are lost (Fig. 2b). Despite substantial refolding of the penton-VP5 dimerization domain in KSHV, P6 has a dimerization domain that is flexible, rather than one that forms a helix-pair with Pen1 as observed in α-herpesvirus (Fig. 2b). Perhaps correlated with the structural reorganization at the dimerization domain, the N-lasso of P1 as well as penton-VP5s refolds from a long, extended lasso structure (observed in P6 and typical-hexon VP5s) into a short β-hairpin, establishing new quasi-equivalent twofold interactions above the helix-pair in α-herpesviruses (Fig. 2b). While the N-lasso of either hexon-VP5 or penton-VP5 in β-herpesviruses shares a similar fold, extending out and clasping a pair of VP5s located diagonally across a local quasi-equivalent twofold axis (Fig. 2b). Coincidently, the N-lasso of P1 rather than clasping a pair of penton-VP5s, refolds into a configuration that largely eliminates its lassoing ability in KSHV. Meanwhile, the N-lasso of penton-VP5s is mostly disordered, which loses the lasso to the P1–P6 pair, further decreasing the interactions between the penton and surrounding P-Hexs (Fig. 2b). In summary, HSVs possess subtle but profound structural differences when compared to β- and γ-herpesviruses. A structure-based phylogenetic analysis suggests that γ-herpesviruses occupy a position that potentially bridges α-with β-herpesviruses and seems slightly close to α-herpesviruses (Supplementary Fig. 5), which is in line with some common biological features of HSVs, including similar genome size and presence of CVSC.

**Structure of the CVSC**. The vertex of the capsid of herpesvirus is a hub for protein–protein interactions that are important for DNA packaging, maturation of the capsid, and egress of the capsid from the infected cells[30]. In HSV-2 C-capsids, five copies of CVSC density are arrayed around the exterior of each of the capsid vertices bridging two triplexes (Ta and Tc) (Fig. 3a, b). Possibly due to the lack of protection by tegument proteins/envelope, the densities at higher radii (corresponding to the VP5 upper-domain and CVSC) are relatively weaker and less well-defined than those at lower radii (Fig. 3a and Supplementary Fig. 2). The overall resolution for the CVSC map is 4.3 Å. The core body of the CVSC has a resolution better than 4.0 Å, and the resolution of the head region of the map is ~10 Å (Fig. 3a and Supplementary Fig. 2). Unexpectedly, in contrast to the bi-lobed head observed in HSV-1[23], only a single-lobed UL25 is seen lying

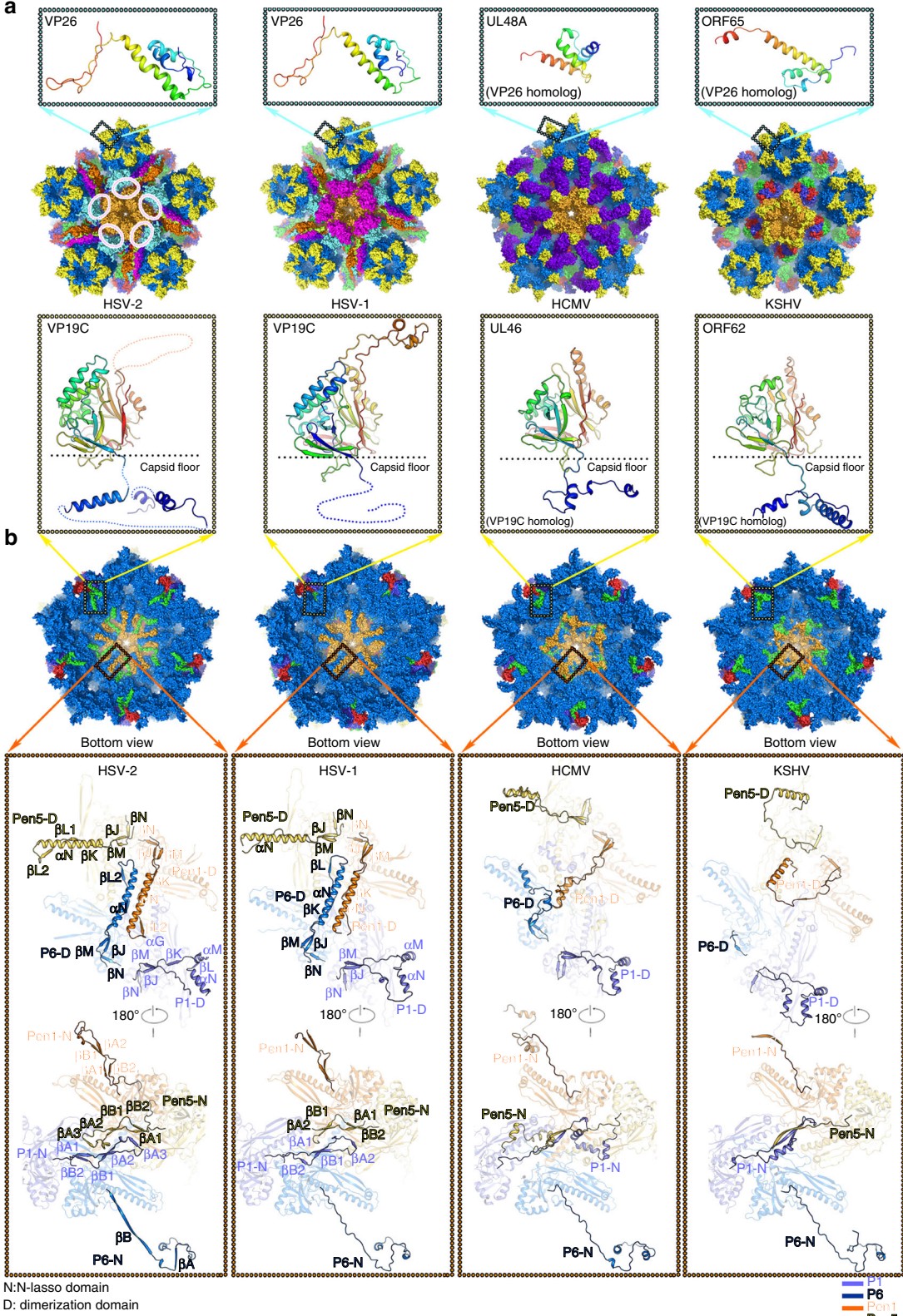

**Fig. 2** Structural comparison of the pentonal vertex of α-, β-, and γ-herpesviruses. Surface representations of the pentonal vertex from α- (HSV-1, HSV-2), β- (HCMV), and γ-herpesviruses (KSHV) viewed from the top (**a**) and bottom (**b**) are shown. The color scheme is same as Fig. 1c, the β-herpesvirus-specific tegument protein pp150 is colored in purple. Red-colored elliptical lines indicate the UL25-1 C domains that were not modeled because of its flexibility in HSV-2. The major structural differences observed in α-, β-, and γ-herpesviruses are marked by cyan (for VP26 and its homologs), yellow (for VP19C and its homologs) and red boxes (for inner capsid around the interface of Pen1, Pen5, P1, and P6). Enlarged views of the structural differences (boxed) are shown with secondary structural elements colored and labeled. For **a** and **b**, rainbow ribbon models show individual proteins (blue N terminus through green and yellow to red C terminus)

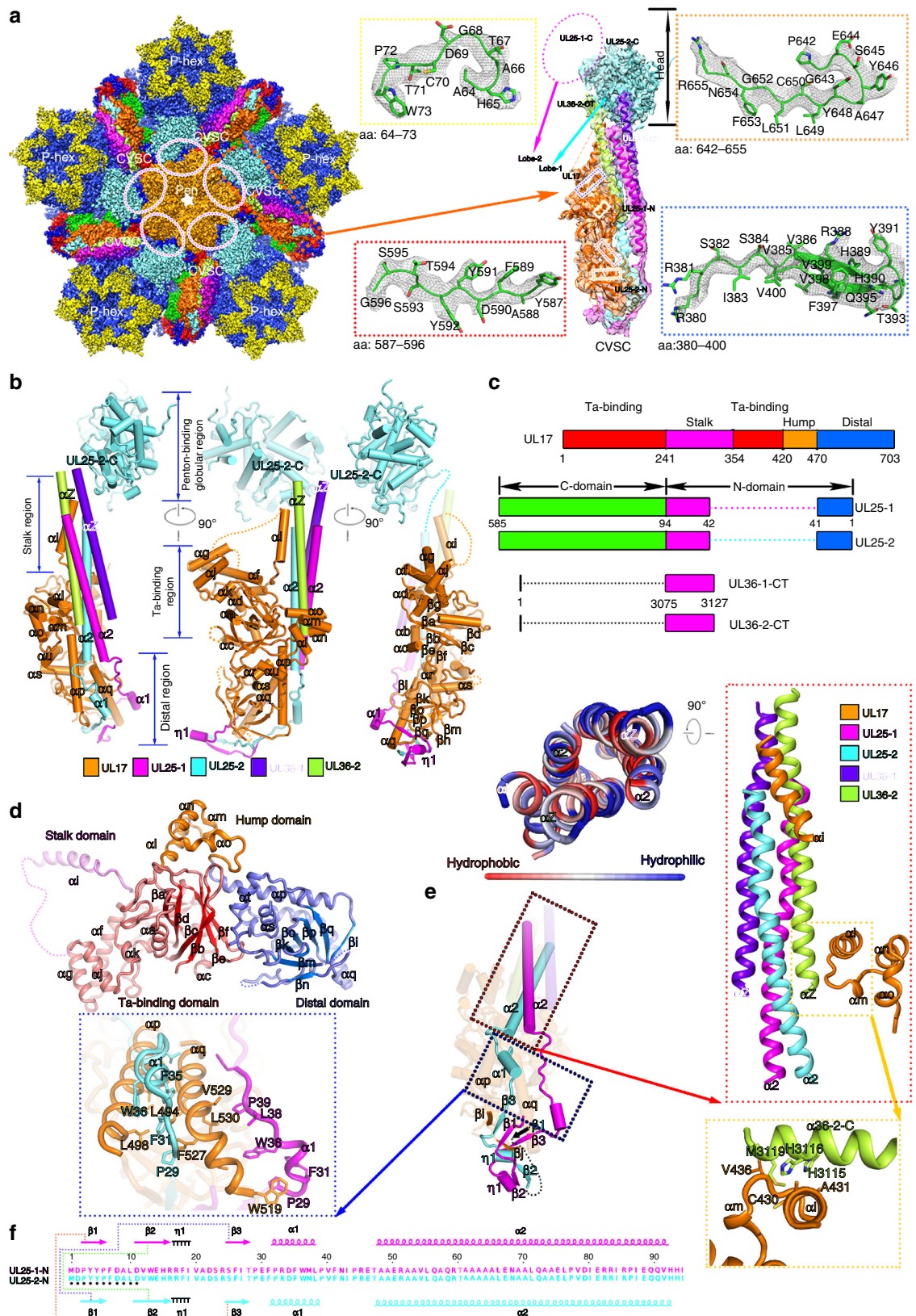

between the space of two adjacent penton-VP5s, indicative of the high level of flexibility or sensitivity to proteases possibly due to the exposure by detergent treatments (Fig. 3a). For the core body, the EM map reveals the polypeptide backbone and many bulky

side chains, allowing atomic modeling (Fig. 3a), and the crystal structure of UL25 C domain (PDB code: 2F5U) can be fitted accurately into the head region of the map, and consistent with observations in the HSV-1 virion[23]. The CVSC core body

**Fig. 3** The CVSC structure. **a** Cryo-EM densities of the CVSC surrounding a pentonal vertex. The color scheme is same as in Fig. 1c. The density for the UL25-1 C domain marked by magenta-colored dashed lines, is not observed due to its flexibility in our map. The inset shows the density map (mesh) and atomic model of CVSC which illustrate side chain features. **b** Pipe-and-plank depictions of the CVSC. The color scheme is same as in Fig. 1c. **c** Schematic diagram of domain organization of the CVSC five components. **d**, **e** Cartoon representation of domain organization of UL17 and two UL25 conformers. Top view of the five-helix bundle (red inset); the left shows the five-helix bundle (colored by residues' hydrophobic characters: hydrophobic to hydrophilic gradient) viewed along the center of the coiled coil. The blue and yellow insets highlighting the hydrophobic interactions and hydrophobic side chains are shown as sticks. **f** represents amino acid sequences of the N domain from two UL25 conformers with secondary structural elements labeled; three sets of small β sheets formed separately by two conformers are indicated by three colored lines. The N-terminal 11 residues which are disordered in the structure of UL25-2 are marked by black dots

comprises one UL17 monomer, two copies of UL25 N domain (residues 1–90) and two copies of UL36 C-terminus (CT, residues 3,072–3,109) while the CVSC head consists of two copies of UL25 C domain (residues 134–580; although one copy is disordered in HSV-2), which are arranged together in a 180-Å-long "gun-like" shape (Fig. 3b).

The CVSC can be functionally divided into four regions: a globular region at the upper end that binds the penton; a stalk region in the middle primarily holding the five components together; a region beneath the stalk region that interacts with Ta, and a distal region that binds Tc and the P-Hex (Fig. 3b). Amongst these, the UL17, the major component of CVSC, involved in the formation of three regions, comprises four domains named as Ta-binding domain, stalk domain, hump domain and distal domain based on their roles in the composition of CVSC (Fig. 3b–d). The two UL25 conformers, engaged in forming three regions of CVSC as well, consist of N and C domains. However, the two N domains of the two UL25 conformers differ in structural details but possess the same secondary structural organization, and can be further divided into two segments: N-extension (residues 1–47) and a coiled-coil helix (α2) (Fig. 3c, e). The penton-binding globular region comprising the C domain of the UL25 attaches itself to the VP5 upper domain (Fig. 3a, b). The Ta-binding region implicated in interaction with Ta consists of Ta-binding domain of the UL17 (residues 1–240 and 354–419), which bears an eight-stranded β barrel and a number of short helices. A five-helix bundle from five components (contributed by αi from the stalk domain of UL17, two α2 helices from two UL25 conformers and two C-terminal helices, α36-C, from two copies of UL36) and a four-helix turn (the hump domain of UL17) make up the stalk region, which bridges the penton-binding and Ta-binding regions (Fig. 3b–c). This structural organization also verifies previous experimental observations that UL36, acting as an integral part of the CVSC, has its C-terminal portion bound to UL25 and the penton[16,31,32]. Extensive hydrophobic interactions from the five-helix bundle and four-helix turn tightly integrate the five components into the CVSC. The distal domain of the UL17 (residues 470–703), harboring a seven-stranded β barrel and a helix-rich section around the β barrel, and the two copies of UL25 N-extension comprise the distal region, which approaches the Tc. At the distal region, the N-extensions of two UL25 conformers (magenta for UL25-1; cyan for UL25-2 in Fig. 3e) intertwine with each other such that three sets of small β sheets (UL25-1 β1 and UL25-2 β3; UL25-1 β2 and UL25-2 β2; UL25-1 β3 and putative UL25-2 β1, albeit that β1 is disorder in UL25-2) are formed by two conformers (Fig. 3e, f). Moreover, β strands (βi and βj) from the distal domain of UL17 join in two sets of two-stranded sheets (UL25-1-β1:UL25-2-β3 and UL25-1-β3:UL25-2-β1) within the UL25 dimer and further establish two sets of three-stranded sheet, integrating the distal domain of UL17 into the CVSC distal region. Additionally, a large hydrophobic patch contributed by αp and αq from UL17 and two α1 helices from the UL25 dimer augments the integration as well (Fig. 3e). In summary, a five-

helix bundle in the stalk region, a large hydrophobic patch and two sets of three-stranded sheets in the distal region drive the assembly of these five components into the CVSC.

**Interactions between CVSC and surrounding capsid proteins.** We note that one CVSC contacts two subunits of the penton, two subunits (P1 and P6) from single P-Hex and two triplexes (Ta and Tc) in an asymmetric manner (Fig. 4). Although densities for one of the di-lobed head sitting atop the penton in HSV-1[23] are not observed in our map, the sole head (UL25 C domain) approaches two VP5 subunits of the adjacent penton via electrostatic attractions (Fig. 4). The Ta-binding region not only contacts the EF and IJ loops from VP23-1 and VP23-2 in Ta triplex, but also attaches its αf to the groove formed by RS- and VW-loops of P6 (Fig. 4). However, neither VP5 subunits (penton- or P-Hex-VP5s) nor triplexes (Ta or Tc) show direct interactions with the stalk region, highlighting the role of the stalk region in intrinsic integration of the CVSC (Fig. 4). The N-extensions of the UL25 dimer insert a number of hydrophobic side chains into two hydrophobic cavities of Tc triplex, one within the VP19C and the other at the interface among the three components of Tc, stapling itself in the Tc (Fig. 4). Additionally, the no and kl loops of the distal domain of UL17 maintain contacts with the VW loop of P1, further fixing the location of the CVSC (Fig. 4 and Supplementary Fig. 6). Overall the interactions with the CVSC contributed equally by two triplexes (Ta and Tc) and P-Hex subunits (P1 and P6) determine the contacts formed with all microenvironments, suggesting that the penton may not be essential for CSVC binding. In line with this, a recent cryo-EM structure of the HSV-1 portal-vertex at sub-nanometer resolution reveals that the portal-vertex CVSC density closely resembles that seen at the penton-vertices[33], suggesting CSVC binding is independent of the penton. Additionally, steric hindrance resulting from differences in the relative orientation, in particular the Ta–Tc pair with its contacting capsomers, prevents the binding of the CVSC to another pair (Tb–Te) that shows the same arrangement as the Ta–Tc pair(Supplementary Fig. 7).

## Discussion

The fact that pentons lose their N-lassoing abilities with surrounding P-Hexs with ~35% less interaction areas compared to hexon–hexon contact areas[22], probably accounts for the vulnerability of pentonal vertices of the herpesvirus capsids to disintegration[34]. The CVSC in HSVs as well as the ORF32 and ORF19 in KSHV[35] or pp150 in HCMV[18] seems to be an adaptation to allow DNA-filled C-capsids to cope with the pressure of its large genome. Five copies of the CVSC together with five copies of Ta and Tc cement the connections between the penton and its five neighboring P-Hexs, firmly stabilizing the pentonal vertex to retain the viral genome (Fig. 3a). The exceptionally high structural and mechanical stability of the C-capsids of HSVs can probably be attributed to the assembly of the CVSC observed in HSV-1 and HSV-2 structures. In fact, CVSC have also been

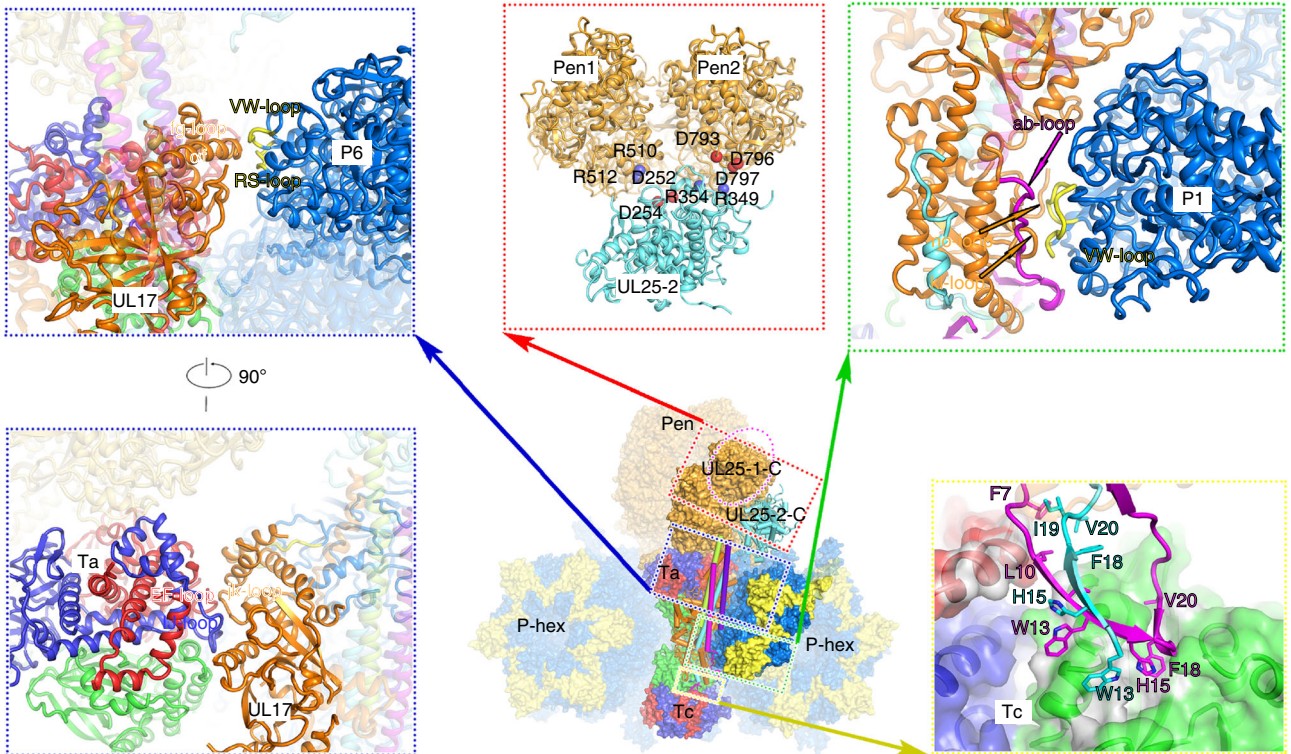

**Fig. 4** Interactions between CVSC and surrounding capsid proteins. The CVSC is shown as cartoon and the remaining structure is rendered as a surface. Capsid proteins contacting the CVSC are highlighted in bright colors and labeled. Model of the UL25-2: Pen–VP5 interface (red inset) highlighting the charge interactions. Interactions between the Ta-binding region of the CVSC and the Ta, P-Hex (blue inset). To show the Ta-binding region clearly, the stalk region is set with 70% transparency. The green and yellow insets show the contacts between the distal region and the Tc, P'-Hex. The color scheme is the same as that of Fig. 1c. The loops in P1 and P6 involved in binding to the CVSC are highlighted in yellow. The N-extensions of two UL25 conformers insert into two cavities of Tc triplex via hydrophobic interactions. Hydrophobic residues from two UL25 conformers and Tc triplex are shown as sticks and presented as gray surface, respectively

shown to be required for structural reinforcements at the vertices, the most stressed part of the capsids[36]. In line with this, thermal stability assays demonstrated that HSV-2 C-capsids can withstand remarkably high temperature (up to 76 °C), revealing HSV-2 C-capsid is one of the most robust viral capsids (Supplementary Fig. 8). Such stability protects capsids from physical damage during long distance transportation involving mechanical forces generated by molecular motors in the axonal cytoplasm before they reach the nuclear pores for releasing the genome.

The near-atomic structure of the HSV-2 C-capsid reported here, together with recent cryo-EM structures of HSV-1, HCMV, and KSHV, expands our understanding of the structural similarities and differences of the α-, β-, and γ-herpesviruses subfamilies, enabling identification of α-herpesvirus-specific structural features. Structure of the CVSC and its interactions with capsid proteins coupled with functional studies suggest multiple potential functions like providing reinforcement for enhancing the stability of the capsid, maturation of capsid, egress of the capsid, initiating primary tegumentation for neonatal virons, and retrograde transport of the incoming viral capsid during initial infections. Our structure provides information on the modes of interaction between the capsid and teguments, which would unveil crucial information on the space–time complexity in the herpesvirus life cycle and may ultimately inform future therapeutic inventions.

## Methods

**HSV-2 capsid purification**. Vero cells without mycoplasma contamination were cultured in Dulbecco's Modified Eagle Medium (DMEM) plus 10% fetal bovine

serum (FBS) and grown to the concentration of $1.5 \times 10^8$, then inoculated with herpes simplex virus 2 strain *MS* at a multiplicity of infection (MOI) of 0.1–1. The herpesvirus-infected cells were scraped from the plates and harvested when reaching 90% cytopathic effect, and resuspended in PBS containing 1% NP-40. Cells were lysed by three cycles of freezing and thawing. After lysis, the suspension was centrifuged at 1500*g* for 15 min at 4 °C to remove cell debris. We further enrich the capsids by a discontinuous 20 and 60% sucrose gradient (w/v in PBS) and collected the band at interface of the two sucrose layers. To separate three types of capsids, crude HSV-2 capsids (~0.6 mg in 600 μl PBS) were loaded onto a continuous 20–50% (w/v in PBS) sucrose density gradient (made by Biocomp) and centrifuged at 80,000*g* for 1 h using a SW40 rotor (Hitachi company). Three sets of fractions were collected and dialyzed against PBS buffer.

**Cryo-EM and data collection**. The cryo-grids were prepared using Thermo Fisher Vitrobot. A 3 μl aliquot of purified HSV-2 C-capsids was applied to a fresh glow-discharged 400-mesh holey carbon-coated copper grid (C-flat, CF-2/1–2C, Protochips). After being blotted for 3.5 s in 80% relative humidity, the grids were plunged into liquid ethane cooled with liquid nitrogen. Cryo-EM datasets were collected at 300 kV with a Titan Krois microscope (FEI) equipped with a GIF Quantum energy filter (Gatan) operated in zero energy-loss mode with a slit width of 20 eV and a direct electron detector (Falcon3). Movies (25 frames, each 0.2 s, total dose 25 e$^-$ Å$^{-2}$) were recorded with a defocus between 0.8 and 2.3 μm using SerialEM[37] at a nominal magnification of 59,000×, which yields a final pixel size of 1.41 Å.

**Image processing**. A total of 10,280 micrographs were recorded. Frames 3–22 were used and corrected for beam-induced drift by aligning and averaging the individual frames of each movie using MOTIONCORR[38]. The contrast transfer function (CTF) parameters for drift-corrected micrographs were estimated by Gctf[39]. 7902 micrographs with visible CTF rings beyond 1/5 Å in their spectra were selected for further processing. Particles were picked automatically by EMAN package[40]. A total of 64,659 particles from 7902 micrographs were firstly picked for the two-dimensional alignment and three-dimensional reconstruction with Relion[25]. The structural flexibility of the capsid and the defocus gradient on this

large capsid limit the resolution, only yielding a reconstruction of 4.2 Å resolution with icosahedral symmetry imposed. To overcome these two problems, we developed a block-based reconstruction[22]. In our case, icosahedral orientation and center parameters of each particle image determined by Relion[25] were used to guide extraction of components of four block regions (Pen-, P-Hex-, E-Hex-, C-Hex-blocks, ~50% bigger than each capsomer) and all four blocks were refined and reconstructed separately with their local mean defoci. For each boxed particle, there were 60, 30, 60, and 12 icosahedral-symmetry-related copies for P-Hex-, E-Hex-, C-Hex-, and Pen-blocks respectively. On condition that we acquired the rotation and translation parameters of a virus calculated by Relion, the distance $d$ between the center of the virus and the center of one copy in the 3D virus along the Z axis can be further calculated to determine the gradient in defocus through the capsid. Assume that the defocus obtained by fitting the Thon ring can represent the distance between focused point of objective lens and the center of the virus. So the local defocus of each copy in the 3D virus was the sum of $d$ and the defocus value. This local defocus of each copy, rather than the uniform defocus obtained by fitting the Thon ring, was used to reconstruct the blocks. After refinement and reconstruction of the four blocks in Relion[25], the resolution of maps for Pen-, P-Hex-, E-Hex-, C-Hex-blocks was further improved to 4.0 Å, 3.75 Å, 3.72 Å, and 3.71 Å respectively, as determined by Golden standard Fourier shell correlation at the 0.143 threshold[41]. A program was used to combine the four blocks into an asymmetric unit. After combining these blocks, the final resolution of HSV-2 C-capsid is 3.75 Å.

**Model building and refinement.** The 3.1 Å structure of the HSV-2 B-capsid[22] was initially fitted into the EM map with CHIMERA[42] and further corrected and adjusted manually by real-space refinement in COOT[43]. The models of the CVSC were built de novo into density using COOT[43]. Models were further improved by iterative positional and B-factor refinement using Phenix[44], rebuilding in COOT[43] and evaluated by Molprobity[45] and Refmac[46]. Refinement statistics are listed in Supplementary Table 1.

## Data availability

The cryo-EM density map of HSV-2 C-capsid has been deposited in the electron microscopy data bank under accession code EMD-6976 and the atomic coordinates of the asymmetric unit have been deposited in the protein data bank under accession code 5ZZ8. Additional data that support the findings of this study are available from the corresponding authors upon reasonable request.

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

### Acknowledgements

We thank Dr. Xiaojun Huang, Dr. Boling Zhu, and Dr. Zhenxi Guo for cryo-EM data collection at the Center for Biological imaging (CBI), Institute of Biophysics. Work was supported by the Strategic Priority Research Program (XDB29010000, XDB08020200), the Key Programs of the Chinese Academy (KJZD-SW-L05), National Key Research and Development Program (2014CB542800, 2017YFC0840300 and 2017YFA0504701), National Science Foundation of China (813300237, 31570717, 91530321, 31570742 and 81520108019) and Technology Planning Project of Hunan Province (No. 2017RS3033) and. X.W. was supported by Young Elite scientist sponsorship by CAST and the program C of "One Hundred of Talented People" of the Chinese Academy of Sciences.

### Author contributions

J.W., S.Y., H.T., N.W., W.C., and X.W. performed the experiments, Q.G., Y.L., and J.W. provided the reagents, D.Z., H.L., X.Z., and X.W solved the structure, and X.W. and Z.R. designed the study. All authors analyzed the data and X.W. and Z.R. wrote the manuscript.

## Additional information

**Competing interests:** The authors declare no competing interests.

