## [Peer Review File · Nature Communications]

Reviewers' comments:

Reviewer #1 (Remarks to the Author):

Nature Communications Manuscript NCOMMS-18-04918-T

Title: The architecture of the Herpes simplex virus C-capsid

Authors: J. Wang, S. Yuan, D. Zhu, H. Tang, N. Wang, W. Chen, Q. Gao, Y. Li, J. Wang, H. Liu, X. Zhang, Z. Rao and X. Wang

The capsids of herpes family viruses are among the largest and most complex of the viruses that infect humans. Capsids are composed of at least 8 distinct protein species, most present in many copies each. The major capsid protein and VP26 homolog proteins, for instance, are each present in nearly 1000 copies, and each of the three triplex proteins are present in 320 copies per capsid. The capsid proteins are linked together by many distinct contacts which investigators have long wanted to target with novel anti-herpes therapeutics. If only atomic-level structures of the capsids were available.

Now they are. Three studies have led the way, structures of the human cytomegalovirus, KSHV and HSV1/2 capsids (refs 8, 9, and the present authors' structure of HSV1 B-capsids). Cryo-electron microscopy was used for each study with modern electron detectors enabling the atomic level resolution that distinguish the present work from prior, lower resolution studies. Work described in the present study adds a structure of the capsid vertex-specific component (CVSC) of HSV2 to the prior atomic level resolution studies. Five CVSC's surround each of the 12 capsid pentons where one of their roles is the mechanical fortification of the penton against the pressure the virus DNA exerts inside the capsid wall.

The CVSC component proteins (UL25, UL17 and a portion of UL36) are linked by distinctive interactions that suggest themselves as targets for therapeutic compounds that would disrupt the CVSC structure. As a result of the resolution of the current reconstruction it is now possible for investigators to design and test appropriate inhibitors. The enhanced resolution compared to prior structures makes all the difference.

I like every aspect of the new structure. It clearly lays out the structure of each CVSC component and how each interacts with the others. Structures similar or identical to the HSV2 CVSC are present in all herpesvirus capsids insuring the current structure will have wide significance. The newly described interactions are themselves somewhat unusual involving alpha helices and regions of beta structure rather than the C-terminal tails often found to link virus capsid proteins. The work is reliably done and described clearly in the text and figures of the paper. The work will be of interest to all herpes virologists and to those newly-attracted to the field by the possibility of identifying novel anti-herpes agents.

The most important weakness of the paper is the fact that much of it is repeated from the authors' Science paper. This applies to all of Supplementary Figure 2 except for figures describing the CVSC (Supplementary Fig. 2h). Supplementary figures 3-6 add little to the overall contribution and can be deleted. I could not get much meaning from Fig. 2. The differences described between the B and C capsids are small and they are not conveyed effectively in the graphics presented. Figures 1 and 3 are wonderful.

Other comments:

1. Line 43: 3.75A resolution
2. Line 52: ...a protective layer for DNA
3. Discussion: The structure determined for UL25. Is it identical to that reported by Bowman et al. in 2006 (J. Virol. 80: 2309-2317)? Clarify.

Reviewer #2 (Remarks to the Author):

NCOMMS-18-04918-T

The architecture of the Herpes simplex virus C-capsid

This is a rather difficult manuscript to review for several reasons. Firstly, it is not clear to me from the submission documents supplied to me whether the 3.1Å structure of a B-type HSV capsid is accepted in Science or just submitted?

From the way that this current submission is written, I assume that it was originally submitted as a letter to Nature? I am not surprised that Nature passed on it, but my overall feeling is that this is a significant advance in structural virology, and I am therefore strongly supportive of its eventual publication in Nature Communications.

However, having said that, in my opinion the manuscript requires work to make it ready for publication. The Nature letter format is very different to that required for Nature Communications, and I will not focus on those format changes – but clearly the manuscript should now be significantly expanded in scale and scope with proper introduction and methods sections. With this in mind, I don't think it's worthwhile to provide a detailed critique of the manuscript in its current form.

More generally, I am not an expert in the structure or biology of the herpesviridae, but I am experienced in both cryo-EM and structural virology – I am therefore a 'somewhat expert' reader and even so found this manuscript incredibly dense and difficult to follow. The authors should make every effort to counter this criticism in their revisions, including I suggest:

1. Clarifying the relationship of A B & C-type capsids, their proportions, and their importance/relevance to the HSV lifecycle.
2. A fuller and slower introduction to the proteins that build the capsid. I understand and appreciate the difficulty of presenting such an enormous structure, but the longer format of Nature Comms gives them much more scope to do their beautiful structure justice.
3. The idea that pressure expands the capsid, is tempting but as far as I can tell completely unjustified? Why can the capsid not undergo a conformational change upon DNA entry? This is mentioned throughout the manuscript, and I found the idea interesting but poorly discussed/justified. Looking at the sup figures, the capsid actually seems to expand most at the vertices – not what I would expect. Is this correct?
4. The FSC plot looks overly smoothed. Has a smoothing function been applied.
5. Some of the figures are v. low resolution – too low to judge their utility. E.g. local resolution in sup
6. I found figure 1 too small, dense and ultimately it should be improved (and made as large as possible!). The colour coding and schematic only partly works, especially in the CVSC that is a key feature of this structure. The figures could now be expanded in a Nature comms article, and the journal should make every effort to reproduce them at the maximum possible size.

I would be very happy to revisit a suitably revised article.

Reviewer #3 (Remarks to the Author):

Wang et al. present a cryo-EM structure of the DNA-filled C-capsid of herpes simplex virus type 2 (HSV-2) at 3.75 Å resolution, which they obtained using a technique they termed block-based reconstruction, described in a companion paper now in press. In this approach, a large particle is

divided into several parts which are reconstructed individually and then recombined, yielding a higher resolution reconstruction of the entire particle. In the companion paper (provided), the authors present the HSV-2 B-capsid at 3.1 Å resolution. While these structures seem to be correct and the resolutions obtained are quite impressive, B- and C-capsids are very similar except for the presence of the capsid-vertex-specific component (CVSC) on the C-capsid. The paper submitted is technically accomplished and brings some new information on an important macromolecular complex: as such it is potentially publishable but I would ask for significant revision in the following directions.

1) The CVSC is a complex consisting of the viral proteins UL17, UL25, and UL36. In this context, a more specific title to this paper would be helpful to focus on it as the C-capsid architecture has long been known. Similarly, the current abstract is not very informative about what this paper brings and should be reconsidered.

2) It is argued that the CVSC stabilizes the capsid, enabling it to withstand the pressure exerted by the packaged genome. The authors argue that C-capsids, when compared to B-capsids, expand. This should be documented quantitatively. The added stability provided by the CVSC, particularly UL25, has been subject of an atomic force microscopy study by Snijder et al. (ref 16). The only stability-related data presented here are in Figure 3d and they make a cursory comparison of HSV-2 with several other viruses which are so different in major respects from HSV-2 as to make this comparison of little value. I advise dropping Figure 3d.

3) Also relating to capsid stability and its ability to resist stress from packaged DNA, a high-resolution structure of the human cytomegalovirus (HCMV) capsid has recently been published (ref. 8). As the HCMV capsid is about the same size as that of HSV-2 and is able to harbor a genome that is 50% larger, this bears some discussion.

4) The main new information concerns the CVSC and this should be more fully documented. What was the occupancy of CVSC per vertex in the preparation(s) of C-capsids analyzed? The description of the CVSC in Supp. Fig. 2 should be transferred into the main paper. It is known that parts of the CVSC are flexible. Accordingly, while local resolution slices are provided in supplementary figure 1b, more focus needs to be put on the resolution of the CVSC in particular: the electron densities shown already suggest to this reviewer that resolution is not uniform across the particle and appears to be significantly lower in the vicinity of the CVSC.

5) It has recently been suggested that in gamma- and alpha-herpesviruses the CVSC contains two copies of UL25, rather than just one (Dai et al. – ref 15). This issue should be addressed. Are the present observations consistent with a UL25 dimer? And if so, which subunit is present on C-capsids?

6) While the resolution obtained in this study is creditable, 3.75 Å (and probably lower around the CVSC – see (4) above) is not 'atomic resolution', as is claimed in line 171. Accordingly, this term should be avoided.

Manuscript Title: " **The architecture of the Herpes simplex virus C-capsid** "

Tracking #: NCOMMS-18-04918-T

Response to referees' comments

We thank the reviewers for their positive and constructive comments, and we believe the paper is now improved.

Reviewers' comments:

Reviewer #1 (Remarks to the Author):

Nature Communications Manuscript NCOMMS-18-04918-T

Title: The architecture of the Herpes simplex virus C-capsid

Authors: J. Wang, S. Yuan, D. Zhu, H. Tang, N. Wang, W. Chen, Q. Gao, Y. Li, J. Wang, H. Liu, X. Zhang, Z. Rao and X. Wang

The capsids of herpes family viruses are among the largest and most complex of the viruses that infect humans. Capsids are composed of at least 8 distinct protein species, most present in many copies each. The major capsid protein and VP26 homolog proteins, for instance, are each present in nearly 1000 copies, and each of the three triplex proteins are present in 320 copies per capsid. The capsid proteins are linked together by many distinct contacts which investigators have long wanted to target with novel anti-herpes therapeutics. If only atomic-level structures of the capsids were available.

Now they are. Three studies have led the way, structures of the human cytomegalovirus, KSHV and HSV1/2 capsids (refs 8, 9, and the present authors' structure of HSV1 B-capsids). Cryo-electron microscopy was used for each study with modern electron detectors enabling the atomic level resolution that distinguish the present work from prior, lower resolution studies. Work described in the present study adds a structure of the capsid vertex-specific component (CVSC) of HSV2 to the prior atomic level resolution studies. Five CVSC's surround each of the 12 capsid pentons where one of their roles is the mechanical fortification of the penton against the pressure the virus DNA exerts inside the capsid wall.

The CVSC component proteins (UL25, UL17 and a portion of UL36) are linked by distinctive interactions that suggest themselves as targets for therapeutic compounds that would disrupt the CVSC structure. As a result of the resolution of the current reconstruction it is now possible for investigators to design and test appropriate inhibitors. The enhanced resolution compared to prior structures makes all the difference.

I like every aspect of the new structure. It clearly lays out the structure of each CVSC

component and how each interacts with the others. Structures similar or identical to the HSV2 CVSC are present in all herpesvirus capsids insuring the current structure will have wide significance. The newly described interactions are themselves somewhat unusual involving alpha helices and regions of beta structure rather than the C-terminal tails often found to link virus capsid proteins. The work is reliably done and described clearly in the text and figures of the paper. The work will be of interest to all herpes virologists and to those newly-attracted to the field by the possibility of identifying novel anti-herpes agents.

We thank the reviewer for a high evaluation of our manuscript and considering it an important contribution.

The most important weakness of the paper is the fact that much of it is repeated from the authors' Science paper. This applies to all of Supplementary Figure 2 except for figures describing the CVSC (Supplementary Fig. 2h). Supplementary figures 3-6 add little to the overall contribution and can be deleted. I could not get much meaning from Fig. 2. The differences described between the B and C capsids are small and they are not conveyed effectively in the graphics presented. Figures 1 and 3 are wonderful.

Thanks for your constructive suggestions. As suggested by you and other reviewers, we have reformatted our manuscript to conform to a Research article, have added more information on structural comparisons with other herpesviruses, which expands our understandings in structural commonness and diversifications of the α -, β - and γ -herpesviruses sub-families. Additionally, the α -herpesvirus-specific structural features are highlighted as well. Most of that described in our Science paper, including a number of Supplementary figures, have been removed in revised manuscript. The Fig.2 in the previous version of the manuscript has been replaced with the structural comparisons of α -, β - and γ -herpesviruses in revised manuscript.

Other comments:

1. Line 43: 3.75A resolution

Thanks, done!

2. Line 52: ...a protective layer for DNA

Thanks, done!

3. Discussion: The structure determined for UL25. Is it identical to that reported by Bowman et al. in 2006 (J. Virol. 80: 2309-2317)? Clarify.

Thanks for pointing this out. We have clarified this in the revised version.

Reviewer #2 (Remarks to the Author):

NCOMMS-18-04918-T

The architecture of the Herpes simplex virus C-capsid

This is a rather difficult manuscript to review for several reasons. Firstly, it is not clear to me from the submission documents supplied to me whether the 3.1Å structure of a B-type HSV capsid is accepted in Science or just submitted?

Sorry, we did not state it clearly - this led to the difficulty in understanding – which was indeed, largely our fault. The 3.1 Å structure of a B-type HSV capsid was published in Science in this April (Ref 23 in the manuscript, Yuan, S. et al. Cryo-EM structure of a herpesvirus capsid at 3.1 Å. Science 360 (2018)).

From the way that this current submission is written, I assume that it was originally submitted as a letter to Nature? I am not surprised that Nature passed on it, but my overall feeling is that this is a significant advance in structural virology, and I am therefore strongly supportive of its eventual publication in Nature Communications.

However, having said that, in my opinion the manuscript requires work to make it ready for publication. The Nature letter format is very different to that required for Nature Communications, and I will not focus on those format changes – but clearly the manuscript should now be significantly expanded in scale and scope with proper introduction and methods sections. With this in mind, I don't think it's worthwhile to provide a detailed critique of the manuscript in its current form.

Firstly, we thank the reviewer for a high evaluation of our manuscript and considering it an important contribution in the field of structural virology. Indeed, our manuscript, originally submitted as a letter to Nature, had a quite different format than that required for Nature Communications. Agreeably, the complex figures needed to accompany the proper introduction and verbose structural descriptions. Therefore, as suggested by Nature Communications as well, we have reformatted our manuscript to an article - this would allow the arguments to be fully presented. In addition, we have made a thorough revision in combinations with your suggestions and comments from other reviewers.

More generally, I am not an expert in the structure or biology of the herpesviridae, but I am experienced in both cryo-EM and structural virology – I am therefore a 'somewhat expert' reader and even so found this manuscript incredibly dense and difficult to follow. The authors should make every effort to counter this criticism in their revisions, including I suggest:

1. Clarifying the relationship of A B & C-type capsids, their proportions, and their importance/relevance to the HSV lifecycle.

Thanks for pointing this out. We have added more background information on A-, B- and C-capsids in the revised manuscript -- as follows. "The assembly pathway produces three distinct types of capsids called A-, B- and C-capsids in lysates from infected cells. A-capsids are empty and result from abortive DNA packing, while B-capsids contain a core comprising of scaffold proteins. Whether B-capsids are abortive forms or assembly intermediates is still debated. The third type of capsid, C-capsid, is fully packed with the DNA genome and matures into an infectious virion."

2. A fuller and slower introduction to the proteins that build the capsid. I understand and appreciate the difficulty of presenting such an enormous structure, but the longer format of Nature Comms gives them much more scope to do their beautiful structure justice.

Thanks for your valuable suggestions. Indeed, a clear presentation of such an enormous structure requires a proper introduction and verbose structural descriptions. As per your suggestions, we have provided a full introduction and detailed descriptions on structure, to present such a complex structure clearly.

3. The idea that pressure expands the capsid, is tempting but as far as I can tell completely unjustified? Why can the capsid not undergo a conformational change upon DNA entry? This is mentioned throughout the manuscript, and I found the idea interesting but poorly discussed/justified. Looking at the sup figures, the capsid actually seems to expand most at the vertices – not what I would expect. Is this correct?

Thanks for pointing this out. It is indeed a good question and reasonable concern. Admittedly, neither experimental evidences nor substantial structural differences support conformational changes of capsid proteins during DNA packaging. We have now recalibrated carefully the pixel size of the C-capsid map using the crystal structure of VP5 upper domain as a benchmark. In doing so, we found out that the fitting was slightly better with the pixel size of 1.41 Å than that with the pixel size of 1.42 Å, suggesting that the observation of the slight expansion in C-capsid might have been caused by the minor deviation in the pixel size. We have corrected our statements on capsid expansion in the revised manuscript-- as follows "The cryo-EM structure of HSV-2 C-capsid reveals an icosahedral assembly of ~ 1,250 Å in diameter, similar to HSV-1, HCMV and KSHV C-capsids and HSV-2 B-capsid (Supplementary Fig. 3). The pressure of tens of atmospheres generated by the packing of the genome does not substantially expand the C-capsid, suggesting C-capsids possess exceptional mechanical and structural stability/rigidity, which is consistent with previous results unveiled with fluid atomic force microscopy."

4. The FSC plot looks overly smoothed. Has a smoothing function been applied.

Thanks for pointing this out. It's our fault that we did not describe clearly the "block reconstruction" procedure. The FSC plot we presented in the manuscript was the mean of those from the four blocks (Pen-, P-Hex-, E-Hex-, C-Hex-blocks, ~50% bigger than each capsomers), which led to an overly smoothed curve. In this revision, we clearly describe the details on how we reconstructed/refined each block and provided the FSC plots for each block, the CVSC and an asymmetric unit.

5. Some of the figures are v. low resolution – too low to judge their utility. E.g. local resolution in sup

Thanks, we are now providing all figures with a high resolution at this stage of review of our work.

6. I found figure 1 too small, dense and ultimately it should be improved (and made as large as possible!). The colour coding and schematic only partly works, especially in the CVSC that is a key feature of this structure. The figures could now be expanded in a Nature comms article, and the journal should make every effort to reproduce them at the maximum possible size.

Thanks for your suggestions. Figure 1 has been expanded to the A4 size (the maximum size) and the color scheme for the CVSC has been optimized as well.

I would be very happy to revisit a suitably revised article.

Reviewer #3 (Remarks to the Author):

Wang et al. present a cryo-EM structure of the DNA-filled C-capsid of herpes simplex virus type 2 (HSV-2) at 3.75 Å resolution, which they obtained using a technique they termed block-based reconstruction, described in a companion paper now in press. In this approach, a large particle is divided into several parts which are reconstructed individually and then recombined, yielding a higher resolution reconstruction of the entire particle. In the companion paper (provided), the authors present the HSV-2 B-capsid at 3.1 Å resolution. While these structures seem to be correct and the resolutions obtained are quite impressive, B- and C-capsids are very similar except for the presence of the capsid-vertex-specific component (CVSC) on the C-capsid. The paper submitted is technically accomplished and brings some new information on an important macromolecular complex: as such it is potentially publishable but I would ask for significant revision in the following directions.

We thank the reviewer for a high evaluation of our manuscript. We have made a careful and thorough revision as per the suggestions of reviewers.

1) The CVSC is a complex consisting of the viral proteins UL17, UL25, and UL36. In this context, a more specific title to this paper would be helpful to focus on it as the C-capsid architecture has long been known. Similarly, the current abstract is not very informative about what this paper brings and should be reconsidered.

Thanks for pointing these out. We have changed the title and have rewritten the abstract to highlight what this paper can bring.

2) It is argued that the CVSC stabilizes the capsid, enabling it to withstand the pressure exerted by the packaged genome. The authors argue that C-capsids, when compared to B-capsids, expand. This should be documented quantitatively. The added stability provided by the CVSC, particularly UL25, has been subject of an atomic force microscopy study by Snijder et al. (ref 16). The only stability-related data presented here are in Figure 3d and they make a cursory comparison of HSV-2 with several other viruses which are so different in major respects from HSV-2 as to make this comparison of little value. I advise dropping Figure 3d.

Thanks for your valuable suggestions. Regarding the expansion of C-capsids, admittedly, neither experimental evidences nor substantial structural differences support conformational changes on capsid proteins during DNA packaging. We have now recalibrated carefully the pixel size of the C-capsid map using the crystal structure of VP5 upper domain as a benchmark. In doing so, we found out that the fitting was slightly better with the pixel size of 1.41 Å than that with the pixel size of 1.42 Å, suggesting that the observation of slight expansion in C-capsid may have been caused by the minor deviation in the pixel size. We have corrected our views on capsid expansion in the revised manuscript. The CVSC forms extensive networks of subunit contacts with surrounding capsid proteins (>2,500 Å²), structurally reinforcing the capsid stability. In line with structural analysis, a number of functional studies have proved the role of the CVSC, in particular UL25, in capsid stability (Ivan Liashkovich et.al, Journal of Cell Science, 2008; Udom Sae-Ueng et.al, Nucleic Acids Research, 2014; Joost Snijder at.al, Journal of Virology, 2017; Wouter H. Roos et.al, PNAS, 2009). As to the thermal stability analysis, agreeably, comparison of HSV-2 with several other viruses adds little value to the manuscript. We have removed this comparison and moved the thermal stability for C-capsids to supplementary materials.

3) Also relating to capsid stability and its ability to resist stress from packaged DNA, a high-resolution structure of the human cytomegalovirus (HCMV) capsid has recently been published (ref. 8). As the HCMV capsid is about the same size as that of HSV-2 and is able to harbor a genome that is 50% larger, this bears some discussion.

Thanks for your suggestions. Yes, we discussed this point in the revised manuscript --as follows “Possibly, to successfully accommodate the 235 kb genome (~50% larger in size than that of HSV) within the similar-size capsid, HCMV evolves HCMV-specific strategies including shortening of its N-anchor of VP19C from 105 to 44 residues. This increases the inner space and squeezes the genome into hexon channels to take full advantage of the inner space”.

4) The main new information concerns the CVSC and this should be more fully documented. What was the occupancy of CVSC per vertex in the preparation(s) of C-capsids analyzed? The description of the CVSC in Supp. Fig. 2 should be transferred into the main paper. It is known that parts of the CVSC are flexible. Accordingly, while local resolution slices are provided in supplementary figure 1b, more focus needs to be put on the resolution of the CVSC in particular: the electron densities shown already suggest to this reviewer that resolution is not uniform across the particle and appears to be significantly lower in the vicinity of the CVSC.

Thanks for your suggestions. We have added more analysis/evaluations on the CVSC, including the occupancy, FSC plot, local resolution evaluation and electron densities in the revised manuscript. The description of the CVSC in Supp. Fig. 2 has been moved to the main paper as well.

5) It has recently been suggested that in gamma- and alpha-herpesviruses the CVSC contains two copies of UL25, rather than just one (Dai et al. – ref 15). This issue should be addressed. Are the present observations consistent with a UL25 dimer? And if so, which subunit is present on C-capsids?

Many many thanks for pointing this out, which raises us a big concern albeit that it was debatable for the existence of one or two copies of UL25 in the CVSC. Dai et al reported that the CVSC contains two copies of UL25 (shaped into a bi-lobed head) in gamma-herpesvirus (KSHV). However, the corresponding densities (for two copies of UL25 in KSHV) in cryo-EM of HSV-1 at 6.8 Å were assigned as one UL25 and one UL36 (Alexis Hue et.al, NSMB, 2016). In contrast to these, only a one-lobed head, not a bi-lobed head is observed in our reconstruction, indicative of the high level of flexibility or sensitivity to proteases possibly due to the exposure by detergent treatments. After careful multiple-rechecks, we have identified two copies of N-domain of UL25 in CVSC body region, despite observing density for only one Core domain (C domain as well) of UL25. We have modified our model of the CVSC and insist that the CVSC consists of two copies of UL25 in HSV-2, which is consistent with the most recent results of HSV-1 in Science.

6) While the resolution obtained in this study is creditable, 3.75 Å (and probably lower around the CVSC – see (4) above) is not ‘atomic resolution’, as is claimed in line 171. Accordingly, this term should be avoided.

Thanks for pointing these out. We have removed the phrase “atomic resolution” from the manuscript.

REVIEWERS' COMMENTS:

Reviewer #2 (Remarks to the Author):

The manuscript is much improved from the original version I saw, and I recommend publication. Much clearer and better presented. I do not think I have any substantive criticisms that remain. The explanation of the structure, is great, as is the block-based reconstruction description.

The manuscript does require a very good proof read. I came across the following as I read the manuscript, but this is not an exhaustive list!

Line 45, "report a cryo-EM" and "containing the capsid specific..."

Line 93, "comprised" rather than "comprising"

Line 113, "in the β -herpesvirus..."

Line 121, "capsids" not capsid

Line 123, "both these structures"

Line 126, "the HSV-2 B-capsid" and "of the HSV-2 C-capsid"

Line 130, "reliable"? One would hope so, but what does this mean? Either remove the word or provide some measure of reliability!

Line 166, "gold standard" not golden

Line 167, might I suggest "complete" rather than "big"

Line 176, "the capsid"

Line 178, "not found in" rather than "from"?

Line 216, "HCMV has probably evolved HCMV-specific" is probably a bit redundant.

Line 190-220 – Figure 2 could do with some panel lettering to make it more navigable.

Line 265, "relatively weaker and less well defined" Relative to what? Less well defined than what?

Line 272, delete the word "clearly"

Line 312, "from UL17 distal domain" would be better as "from the distal domain of UL17" or similar

Line 318, "sheets" not "sheet"?

Line 341, "closely resembles to that" should be "closely resembles that seen"

Line 343, not sure about the colon in this sentence

Line 367, "The near"

Line 368, "understanding"

Reviewer #3 (Remarks to the Author):

This is a revised version of a paper previously reviewed for Nature Communications which describes the structure of the C-capsid (i.e. a DNA-filled capsid isolated from the nuclei of infected cells) of HSV-2. The original round of review recognized the high quality of the cryo-EM analysis but proposed major revisions to minimize repetition of results already reported in the authors' account of the HSV-2 B-capsid and to expand their paper from the minimalist Nature-Letter format to the more accommodating Nature-Communications format. In the present paper, the specific points that I raised have mostly been addressed. However, the paper is barely recognizable as a revision of the original submission. Although the analysis is of high quality and the study is, in principle, well worthy of publication, I found it very hard to read, with long sentences - not always grammatical - and long, seemingly endless, paragraphs. It would also help to make more use of subtitles to introduce and underscore particular points. In particular, there are issues concerning the portion about the CVSC (lines 176-256). For instance, I could not find the red lines said in line 190 to represent flexible portions of the CVSC; and which parts of the Figure are panels a and b (as mentioned in Fig 2 legend)?

In short, I would recommend acceptance subject to the authors' attending to the points listed

above relating to coverage of the CVSC and the minor but not ignorable points listed below. In fact, I would also urge them to commission a professional copy-editing. Otherwise, the paper risks not having as much impact as it deserves. That would be unfortunate as the authors deserve full credit for their accomplishment.

Some additional points.

Line 130. Omit "reliable" as this is a subjective, self-congratulatory, term.

Line 165. The term is Gold Standard.

Line 171 and many other places. VP19c has a lower case "c".

Line 192. "quite similar" is too vague. Give numbers derived from the ratio of the maximum density above background of the feature of interest relative to the maximum capsid density, and assuming the latter density to reflect 100% occupancy.

Line 192, 193. The plural of "vertex" is "vertices".

Line 200. Delete "across".

Line 201 and numerous other places. Instead of HSV, specify HSV-1 or HSV-2 as appropriate.

Line 268-69. I suggest adding arrows of other markers to Figure 3a to designate what they call the "head" and the "lobes".

Line 275. Change "HSV-1" to "the HSV-1 virion". Ref. 18 dealt with the virion, not a capsid.

Lines 315-6. Refer to a hydrophobic patch contributed by helices aq and ar. However, the latter is not labeled in the referenced Figure 3e.

Lines 334-336. The authors mention mn and jk loops of UL17 and refer to Supplementary Figure 6. However, in that figure, no loops are labeled.

Ref 15. A more appropriate reference for the 26Å spacing associated with packaged DNA in the HSV-1 C-capsid is Booy et al., Cell 64, 1007-1015, 1991.

Manuscript Title: " **The architecture of the Herpes simplex virus C-capsid** "

Tracking #: NCOMMS-18-04918-A

Response to referees' comments

We thank the reviewers for their positive and constructive comments, and we believe the paper is now improved.

REVIEWERS' COMMENTS:

Reviewer #2 (Remarks to the Author):

The manuscript is much improved from the original version I saw, and I recommend publication. Much clearer and better presented. I do not think I have any substantive criticisms that remain. The explanation of the structure, is great, as is the block-based reconstruction description.

We thank the reviewer for the comments, which helped improve the manuscript.

The manuscript does require a very good proof read. I came across the following as I read the manuscript, but this is not an exhaustive list!

Line 45, "report a cryo-EM" and "containing the capsid specific..."

Thanks, done!

Line 93, "comprised" rather than "comprising"

Thanks, done!

Line 113, "in the β -herpesvirus..."

Thanks, done!

Line 121, "capsids" not capsid

Thanks, done!

Line 123, "both these structures"

Thanks, done!

Line 126, "the HSV-2 B-capsid" and "of the HSV-2 C-capsid"

Thanks, done!

Line 130, “reliable”? One would hope so, but what does this mean? Either remove the word or provide some measure of reliability!

Thanks for your suggestion! We have removed the word “reliable”.

Line 166, “gold standard” not golden

Thanks, done!

Line 167, might I suggest “complete” rather than “big”

Thanks, done!

Line 176, “the capsid”

Thanks, done!

Line 178, “not found in” rather than “from”?

Thanks, done!

Line 216, “HCMV has probably evolved HCMV-specific” is probably a bit redundant.

Thanks, done!

Line 190-220 – Figure 2 could do with some panel lettering to make it more navigable.

Thanks for your suggestions. Figure 2 has been split into 2 panels now.

Line 265, “relatively weaker and less well defined” Relative to what? Less well defined than what?

Thanks for pointing it out. We have modified our statement as follows- “Possibly due to the lack of protection by tegument proteins/envelope, the densities at higher radii (corresponding to the VP5 upper-domain and CVSC) are relatively weaker and less well defined than those at lower radii.”

Line 272, delete the word “clearly”

Thanks, done!

Line 312, “from UL17 distal domain” would be better as “from the distal domain of UL17” or similar

Thanks, done!

Line 318, “sheets” not “sheet”?

Thanks, corrected!

Line 341, “closely resembles to that” should be “closely resembles that seen”

Thanks, corrected!

Line 343, not sure about the colon in this sentence

Thanks for pointing it out, we have removed the colon.

Line 367, “The near”

Thanks, done!

Line 368, “understanding”

Thanks, done!

Reviewer #3 (Remarks to the Author):

This is a revised version of a paper previously reviewed for Nature Communications which describes the structure of the C-capsid (i.e. a DNA-filled capsid isolated from the nuclei of infected cells) of HSV-2. The original round of review recognized the high quality of the cryo-EM analysis but proposed major revisions to minimize repetition of results already reported in the authors’ account of the HSV-2 B-capsid and to expand their paper from the minimalist Nature-Letter format to the more accommodating Nature-Communications format. In the present paper, the specific points that I raised have mostly been addressed. However, the paper is barely recognizable as a revision of the original submission. Although the analysis is of high quality and the study is, in principle, well worthy of publication, I found it very hard to read, with long sentences - not always grammatical - and long, seemingly endless, paragraphs. It would also help to make more use of subtitles to introduce and underscore particular points. In particular, there are issues concerning the portion about the CVSC (lines 176-256). For instance, I could not find the red lines said in line 190 to represent flexible portions of the CVSC; and which parts of the Figure are panels a and b (as mentioned in Fig 2 legend)?

We appreciate the reviewer for pointing these out and we have further improved the manuscript taking into consideration the suggestions of the reviewer. Firstly we have modified our statements/descriptions with short and easy sentences and copy-edited our manuscript carefully. Secondly we have split the Figure 2 into two panels with proper labels to make it more navigable. These have made the manuscript easier for the readers to comprehend.

In short, I would recommend acceptance subject to the authors’ attending to the points listed above relating to coverage of the CVSC and the minor but not ignorable points

listed below. In fact, I would also urge them to commission a professional copy-editing. Otherwise, the paper risks not having as much impact as it deserves. That would be unfortunate as the authors deserve full credit for their accomplishment.

Thanks for your suggestions. After a professional copy-editing, we believe the final revised manuscript is further improved.

Some additional points.

Line 130. Omit “reliable” as this is a subjective, self-congratulatory, term.

Thanks for pointing it out, we have removed the word “reliable”.

Line 165. The term is Gold Standard.

Thanks, corrected!

Line 171 and many other places. VP19c has a lower case “c”.

Thanks, corrected!

Line 192. “quite similar” is too vague. Give numbers derived from the ratio of the maximum density above background of the feature of interest relative to the maximum capsid density, and assuming the latter density to reflect 100% occupancy.

Thanks for your suggestion, done.

Line 192, 193. The plural of “vertex” is “vertices”.

Thanks, done!

Line 200. Delete “across”.

Thanks, done!

Line 201 and numerous other places. Instead of HSV, specify HSV-1 or HSV-2 as appropriate.

Thanks, done!

Line 268-69. I suggest adding arrows of other markers to Figure 3a to designate what they call the “head” and the “lobes”.

Thanks, done!

Line 275. Change “HSV-1” to “the HSV-1 virion”. Ref. 18 dealt with the virion, not a capsid.

Thanks, done!

Lines 315-6. Refer to a hydrophobic patch contributed by helices α_q and α_r . However, the latter is not labeled in the referenced Figure 3e.

Thanks for pointing it out, we have labeled them in the Figure 3e.

Lines 334-336. The authors mention mn and jk loops of UL17 and refer to Supplementary Figure 6. However, in that figure, no loops are labeled.

Thanks for pointing it out, we have labeled them in the Supplementary Figure 6.

Ref 15. A more appropriate reference for the 26Å spacing associated with packaged DNA in the HSV-1 C-capsid is Booy et al., Cell 64, 1007-1015, 1991.

Thanks, done!